# Rapid Identification of Constituents in *Cephalanthus tetrandrus* (Roxb.) Ridsd. et Badh. F. Using UHPLC-Q-Exactive Orbitrap Mass Spectrometry

**DOI:** 10.3390/molecules27134038

**Published:** 2022-06-23

**Authors:** Su-Nv Tang, Jian-Bo Yang, Shuai E, Shuo He, Jia-Xin Li, Kai-Quan Yu, Min Zhang, Qing Li, Lei Sun, Hui Li

**Affiliations:** 1School of Pharmaceutical Sciences, Hunan University of Medicine, Huaihua 418000, China; tangsunvxiaoxiannv@163.com (S.-N.T.); eshuai1999@163.com (S.E.); liaoning1143569786@163.com (S.H.); ljx18273525098@163.com (J.-X.L.); kai2020103020111@163.com (K.-Q.Y.); zm18692378021@163.com (M.Z.); lq13319661296@163.com (Q.L.); 2National Institutes for Food and Drug Control, Beijing 100050, China; yangjianbo@nifdc.org.cn; 3School of Pharmacy, Weifang Medical University, Weifang 261000, China

**Keywords:** *Cephalanthus tetrandrus* (Roxb.) Ridsd. et Badh. F., UHPLC-Q-exactive orbitrap MS, neutral loss, diagnostic fragmentation ions, chlorogenic acid derivatives, flavonoids, anthocyanin

## Abstract

*Cephalanthus tetrandrus* (Roxb.) Ridsd. et Badh. F. (CT) belongs to the Rubiaceae family. Its dried leaves are widely used in traditional Chinese medicine to treat enteritis, dysentery, toothache, furuncles, swelling, traumatic injury, fracture, bleeding, and scalding. In order to further clarify the unknown chemical composition of CT, a rapid strategy based on UHPLC-Q-exactive orbitrap was established for this analysis using a Thermo Scientific Hypersil GOLD^TM^ aQ (100 mm × 2.1 mm, 1.9 µm) chromatographic column. The mobile phase was 0.1% formic acid water–acetonitrile, with a flow rate of 0.3 mL/min and injection volume of 2 µL; for mass spectrometry, an ESI ion source in positive and negative ion monitoring modes was adopted. A total of 135 chemicals comprising 67 chlorogenic acid derivatives, 48 flavonoids, and 20 anthocyanin derivatives were identified by comparing the mass spectrum information with standard substances, public databases, and the literature, which were all discovered for the first time in this plant. This result broadly expands the chemical composition of CT, which will contribute to understanding of its effectiveness and enable quality control.

## 1. Introduction

*Cephalanthus tetrandrus* (Roxb.) Ridsd. et Badh. F. (CT), known as Ma Yanshu or Water Yangmei in traditional Chinese medicine (TCM) and ‘Bagua Maple’ in Dong medicine, belongs to the Rubiaceae family. The leaves have the effects of clearing away heat and toxic materials, as well as dispelling blood stasis and reducing swelling, and they are used for the treatment of enteritis, dysentery, toothache caused by acute gingivitis and acute pulpitis, furuncles, swelling, traumatic injury, fracture, bleeding, and scalding [1]. This plant is mainly distributed in Hunan, Guangdong, Hainan, and Taiwan provinces. However, no detailed studies on the material basis of its medicinal effects have been reported. To date, most studies on CT have focused on its ornamental value. A previous study reported on its pharmacognosy [2]. Therefore, it is necessary to clarify the unknown chemical composition of CT.

In order to further analyze and discover its chemical composition and pharmacological effects, it is necessary to find a suitable analytical technique to analyze the complex chemical composition in CT. There are many methods for analyzing herbal medicine, including thin-layer chromatography (TLC) [3,4], ultraviolet spectroscopy (UV) [5], infrared spectroscopy (IR) [6], and nuclear magnetic resonance (NMR) [7,8]; however, the components of Chinese herbal medicine are complex and in trace amounts, and these methods cannot accurately characterize them. UV is only applicable to the determination of groups containing unsaturated bonds and aromatic structures, with low quantitative sensitivity and a small application range. NMR also suffers from low sensitivity, with a narrow detection range. TLC and IR easily succumb to interference by many factors, leading to large errors in the analysis results. In recent years, the combination of liquid chromatography and mass spectrometry has resulted in improvements in sensitivity and resolution, and it has been widely used for the analysis of food, environment, drugs, etc., as well as for the qualitative analysis of the components of TCM [9,10,11].

In this study, a strategy based on UHPLC-Q-exactive orbitrap was established for the identification of the unknown chemical components of CT. As a result, a total of 135 chemicals comprising 67 chlorogenic acid derivatives (CGAs), 48 flavonoids, and 20 anthocyanin derivatives, were identified by comparing the mass spectral information with standard substances, public databases, and the literature, all of which are reported for the first time.

## 2. Results and Discussion

### 2.1. Scheme for Qualitative Analysis

In this study, a strategy based on UHPLC-Q-exactive orbitrap MS combined with parallel reaction monitoring (PRM), diagnostic fragment ions (DFIs), and neutral loss (NL) was established. First, the total extract was obtained via reflux filtration and rotary evaporation. Secondly, the sample was injected into the UHPLC-Q-exactive orbitrap MS to obtain a high-resolution mass spectrum through a full scan. Thirdly, the mass spectral fragmentation pathway library of each chemical component was established and summarized. Fourthly, the potential chemical was predicted by metabolite workflow in Compound Discoverer 3.0 using the following parameters: the drugs were set as shikimic acid, quinic acid, quercetin, and kaempferol, whereas the groups added were set as a list of the abovementioned substituents. Next, the fragment ions were acquired using UHPLC-Q-exactive orbitrap MS in PRM mode triggered by the list of included ions. Fifthly, a high-resolution extraction ion flow diagram (HREIC) was used to further verify the accuracy of screening. Lastly, the candidate chemicals were identified on the basis of diagnostic fragment ions, neutral loss, and retention time, as well as through comparison with the literature.

### 2.2. Optimization of Extraction Conditions

Different extraction conditions, including extraction methods (reflux extraction and ultrasonic extraction) and extraction solvents (70% ethanol, 20% methanol, 40% methanol, 60% methanol, 80% methanol, and 100% methanol) were investigated in our study. Eventually, ultrasonic extraction with 70% ethanol at room temperature was chosen as the optimal conditions according to the numbers and intensity of the peak in UHPLC-Q-exactive orbitrap MS.

### 2.3. UHPLC-ESI-MS^2^ Qualitative Analysis of CGAs and Flavonoids

A total of 135 chemical were tentatively identified by UHPLC-Q-exactive orbitrap MS, comprising 67 CGAs, 20 anthocyanins, and 48 flavonoid derivatives. The chromatographic and mass data of those detected constituents are listed in Table 1, and the high-resolution extracted ion chromatograms (HREICs) are shown in Figure 1.

#### 2.3.1. Identification Based on Reference Standard

Compounds 6, 24, 25, 75, 78, and 104 were observed at 3.41, 4.89, 5.03, 8.06, 8.34, and 10.60 min, corresponding to *trans*-3-caffeoylquinic acid (CQA), *trans*-4-CQA, *trans*-5-CQA, 3,4-dicaffeoylquinic acid (DiCQA), 3,5-DiCQA, and 4,5-DiCQA, respectively, by comparing the retention time and MS data with reference standards.

Compounds 70, 71, 80, 106, 117, 126, and 135 were identified as rutin, isoquercitrin, kaempferol-3-*O*-rutinoside, eriodictyol, quercetin, naringenin, and kaempferol, respectively, by comparing the retention time and MS/MS data with reference standards.

Likewise, compounds 13 and 32 were confirmed as procyanidin B1 and procyanidin B2.

#### 2.3.2. Identification of Speculative Chlorogenic Acid Derivatives

##### Identification of Monoacyl-Quinic Acids and Monoacyl-Shikimic Acids

Compounds 14, 28, 42, and 52 were observed at 4.29, 5.25, 5.79, and 6.34 min, respectively, with the same molecular ion [M − H]^−^ at *m*/*z* 337.0929 (C_16_H_18_O_8_). According to the literature [12,13], they were identified as *trans*-3-*O*-*p*-coumaroylquinic acid (pCoQA), *trans*-4-pCoQA, *cis*-4-pCoQA, and *cis*-5-pCoQA, respectively, according to their retention time and base peak ion in the MS^2^ spectrum.

Compounds 27, 30, 54, and 57 were observed at 5.21, 5.30, 6.39, and 6.57 min, respectively, with the same molecular ion [M − H]^−^ at *m*/*z* 397.1140 (C_18_H_22_O_10_) and fragment ions [M − H]^−^ at *m*/*z* 173.0446 and 191.0553; these compounds were identified as sinapoylquinic acids (SQAs). According to their base peak ion and retention time, they were identified as *trans*-3-SQA, *cis*-3-SQA, *trans*-5-SQA, and *cis*-5-SQA through comparison with the literature data [14].

Compounds 33 and 45 were observed at 5.44 and 5.99 min, respectively, with the quasi-molecular ion [M − H]^−^ at *m*/*z* 353.0878 (C_16_H_18_O_9_), exhibiting identical MS^1^ data and similar MS^2^ data to *trans*-3-CQA, *trans*-4-CQA, and *trans*-5-CQA. Accordingly, compounds 33 and 45 were inferred as caffeoylquinic acids (CQAs). According to their retention time and MS^2^ data, as well as the literature [15,16], compounds 33 and 45 were identified as *cis*-5-CQA and *cis*-3-CQA, respectively.

Compounds 38, 43, 49, 53, and 56 were observed at 5.67, 5.85, 6.17, 6.37, and 6.49 min, respectively, with the quasi-molecular ion [M − H]^−^ at *m*/*z* 335.0772 (C_16_H_16_O_8_), inferring that they may be caffeoylquinic acid lactones (CQLs) or caffeoyl shikimic acids (CSAs). The fragmentation ion at *m*/*z* 161.0233 yielded by quinic acid lactones is characteristic of CQLs [17,18,19,20,21]. Hence, compounds 38, 53, and 56 were identified as 3-CQL, 1-COL, and 4-CQL, respectively, on the basis of their retention time and MS^2^ spectra. Furthermore, compounds 43 and 49 were identified as 5-CSA and 4-CSA, respectively.

Compounds 50, 55, and 59 possessed a deprotonated ion at *m*/*z* 367.1035 (C_17_H_20_O_9_), suggesting that they could be feruloyl quinic acids (FQAs). 4-FQA yielded the base peak ion at *m*/*z* 173.0445, whereas cis-5-FQA and trans-5-FQA yielded the base peak ion at *m*/*z* 191.0554. The configuration of cis or trans was determined from the intensity of these peaks [13,15]. Therefore, compounds 50, 55, and 59 were identified as 4-FQA, trans-5-FQA, and cis-5-FQA.

Compounds 9, 15, 20, 22, 31, and 34 were eluted at 3.81, 4.31, 4.50, 4.74, 5.31, and 5.52 min, respectively, with the quasi-molecular ion [M − H]^−^ at *m*/*z* 515.1406 (C_22_H_28_O_14_) and their fragment ions [M − H]^−^ at *m*/*z* 191.0554, 161.0235, 173.0447, and 353.1092, respectively, indicating that they possessed a CQA moiety. Considering the neutral loss of 162 Da, they were identified as CQA-hexoside [11]. Likewise, compounds 10 and 23 were identified as 4-pCoQA-hexoside and 5-pCoQA-hexoside, respectively [17,18,19]; compounds 16, 29, and 40 were tentatively characterized as 5-FQA-hexoside; compounds 17 and 44 were identified as CQA-dihexoside; compound 35 was identified as CSA-hexoside; compound 41 was identified as CQL-hexoside; and compound 47 was identified as 4-FQA-hexoside [17,18,19].

##### Identification of Diacyl-Quinic Acids and Diacyl-Shikimic Acids

Compounds 61, 91, and 107 were obtained at 6.83, 9.15, and 10.96 min, respectively, with the quasi-molecular ion [M − H]^−^ at *m*/*z* 515.1195 (C_25_H_24_O_12_), indicating that they were dicaffeoylquinic acids (DiCQAs). Compound 91 yielded the base peak ion [M − H]^−^ at *m*/*z* 173.0445 and fragmentation ions at *m*/*z* 179.0340 and 191.0552, suggesting that it might be 1,4-DiCQA [22,23]. Compounds 110, 115, and 118 with the quasi-molecular ion [M − H]^−^ at *m*/*z* 497.1089 (C_25_H_22_O_11_) were characterized as dicaffeoylquinic acid lactones (DiCQLs) or dicaffeoyl shikimic acids (DiCSAs) on the basis of their fragmentation ions. Hence, compounds 110, 115, and 118 were identified as DiCSA, DiCSA, and DiCQL, respectively.

Compounds 97, 99, 112, and 122 were eluted at 9.46, 9.76, 11.34, and 12.67 min, respectively. The quasi-molecular ion [M − H]^−^ at *m*/*z* 499.1246 (C_25_H_24_O_11_) and the fragment ions [M − H]^−^ at *m*/*z* 173.0441, 163.0389, 179.0337, 191.0551, and 135.0436 were consistent with a coumaroyl caffeoylquinic acid (pCoCQA) moiety. The absence of a base peak ion at *m*/*z* 173.0337 of compound 99 was consistent with 3C,5-pCoQA. Likewise, compounds 97, 112, and 122 were identified as 4-pCo,5CQA, *trans*-4-pCo,5CQA, and *cis*-4-pCo,5CQA, respectively. Compounds 101, 102, 103, 111, and 114, with the quasi-molecular ion [M − H]^−^ at *m*/*z* 529.1351 (C_26_H_26_O_12_) and the fragment ions [M − H]^−^ at *m*/*z* 173.0445 or 193.0496 were consistent with caffeoyl feruloylquinic acids (CFQAs). According to the MS^2^ data and retention times reported in the literature [13,22,24], compounds 101, 102, 103, 111, and 114 were identified as 3C,5FQA, 3F,5CQA, 3C,5FQA, 4C,5FQA, and 4F,5CQA, respectively.

##### Identification of Triacyl-Quinic Acids and Triacyl-Shikimic Acids

Compound 127 possessed a molecular ion [M − H]^−^ at *m*/*z* 677.1512 (C_34_H_30_O_15_) and a fragment ion at *m*/*z* 353.0878, consistent with CQA. According to [22], compound 127 was identified as TriCQA.

##### Others

Compounds 1 and 2 were obtained at 0.93 and 1.11 min, with the molecular ion [M − H]^−^ at *m*/*z* 353.1089 (C_13_H_22_O_11_) and fragment ions [M − H]^−^ at *m*/*z* 173.0443, 191.0551, and 179.0549, indicating a quinic acid moiety. Considering the neutral loss of 162 Da, they could be considered as hexosides of quinic acid (QA-hexosides).

Compounds 3, 4, 5, 7, 8, 11, 12, 18, and 21 yielded a deprotonated ion [M − H]^−^ at *m*/*z* 341.0878 (C_15_H_18_O_9_). Considering the neutral loss of 162 Da and the base peaks at *m*/*z* 135.0438 and 179.0339, they were identified as caffeic acids [25]. Hence, they might be considered as hexosides of caffeic acid (CA-hexosides).

#### 2.3.3. Identification of Speculative Anthocyanins

The common anthocyanins in *Cephalanthus tetrandrus* (Roxb.) Ridsd. et Badh. F. are the glycosylated derivatives of pelargonidin (*m*/*z* 271.0601), cyanidin (*m*/*z* 287.0550), peonidin (*m*/*z* 301.0707), delphinidin (*m*/*z* 303.0499), and petunidin (*m*/*z* 317.0667).

Compounds 73 and 79, eluted at 7.98 and 8.38 min, possessed a similar molecular ion [M + H]^+^ at *m*/*z* 595.1657 (C_27_H_31_O_15_) and a fragment ion at *m*/*z* 287.0544 [M − 308.110]^+^, indicating the loss of one rutinose moiety. According to [26], compounds 73 and 78 were determined to be cyanidin-3-*O*-rutinoside isomers.

Compounds 26 and 48 were eluted at 5.14 and 6.02 min, respectively, with the quasi-molecular ion [M + H]^+^ at *m*/*z* 757.2186 (C_33_H_41_O_20_) and fragment ions [M + H]^+^ at *m*/*z* 287.0534, 449.1057, and 595.1647, indicating that they possessed a cyanidin moiety. Considering the neutral loss of 162 Da (757.2186 − 595.1647) and 308 Da (757.2186 − 449.1057), they were identified as cyanidin-*O*-rutinoside-*O*-galactoside [26,27].

Compound 69 was obtained at 7.51 min, with the quasi-molecular ion [M + H]^+^ at *m*/*z* 465.1028 (C_21_H_21_O_12_) and the fragment ion [M + H]^+^ at *m*/*z* 303.0494, indicating the neutral loss of 162 Da; hence, compound 69 corresponded to delphinidin-3-*O*-hexoside. Likewise, compounds 84, 89, 90, and 95 were determined to be petunidin-3-*O*-hexoside, petunidin-3-*O*-galactoside, petunidin-3-*O*-glucoside, and peonidin-3-*O*-hexoside, respectively [26].

Compounds 39 and 87 appeared at 5.68 and 8.68 min, respectively, with an identical molecular ion at *m*/*z* 289.0718 (C_15_H_14_O_6_) in negative mode and 291.0863 (C_15_H_14_O_6_) in positive mode, in addition to negative fragment ions at *m*/*z* 109.0282, 203.0704, and 245.0815 and positive fragment ions at *m*/*z* 139.0388, 123.0440, and 147.0439, in accordance with catechin and epicatechin [28,29].

Compound 64 possessed a molecular ion [M + H]^+^ at *m*/*z* 597.1450 (C_26_H_29_O_16_) and a fragment ion at *m*/*z* 303.0493 [M − 294.094]^+^, resulting from the loss of a xylosyl glucoside, which is characteristic of delphinidin-3-xylosylglucoside.

Compound 67 possessed a molecular ion [M + H]^+^ at *m*/*z* 449.1078 (C_21_H_21_O_11_) and an MS/MS fragment ion at *m*/*z* 303.0495 [M − 146.057]^+^, corresponding to the loss of a rhamnose moiety. According to [26], compound 67 was determined to be delphinidin-3-*O*-rhamnoside.

Compound 68 was eluted at 7.50 min, with the quasi-molecular ion at [M + H]^+^ at *m*/*z* 611.1607 (C_27_H_31_O_16_) and the fragment ion [M + H]^+^ at *m*/*z* 303.0493, indicating the presence of a delphinidin moiety. Considering the neutral loss of 162 Da, compound 68 was identified as delphinidin-3-*O*-rutinoside [26]. Likewise, compounds 82, 86, 93, and 98 were identified as petunidin-3-*O*-rutinoside, petunidin-3-*O*-rutinoside, peonidin-3-*O*-rutinoside, and pelargonidin-3-*O*-rutinoside, respectively [26].

#### 2.3.4. Identification of Speculative Flavonoids

##### Identification of Flavonols

Compounds 36, 58, and 63 were eluted at 5.54, 6.63, and 6.89 min, respectively, with the quasi-molecular ion [M − H]^−^ at *m*/*z* 771.1989 (C_33_H_40_O_21_) and fragment ions [M − H]^−^ at *m*/*z* 301.0349, 463.0898, and 609.1459, respectively, indicating that they possessed a quercetin moiety. Considering the neutral loss of 162 Da (771.1989 − 609.1459) and 308 Da (771.1989 − 463.0898), they were identified as quercetin-*O*-glucosyl rutinoside. Similarly, the quasi-molecular ion [M − H]^−^ at *m*/*z* 755.2040 (C_33_H_40_O_20_) of compound 60, with the loss of 454 Da (755.2040 − 301.0391), could be quercetin rhamnosyl rutinoside. The quasi-molecular ion [M − H]^−^ at *m*/*z* 755.1829 (C_36_H_36_O_18_) of compounds 108, 113, 116, and 119, with fragment ions [M − H]^−^ at *m*/*z* 301.0351 and 609.1461, as well as a neutral loss of 146 Da (755.1829 − 609.1461), were identified as quercetin coumaroyl rutinoside. Compound 62 showed a quasi-molecular ion at *m*/*z* 625.1410 (C_27_H_30_O_17_); considering a neutral loss of 162 Da (625.1410 − 463.0883), it could be quercetin-*O*-sophoroside. Compound 65 was found at 7.24 min, with the quasi-molecular ion [M − H]^−^ at *m*/*z* 739.2091 (C_33_H_40_O_19_) and the fragment ion [M − H]^−^ at *m*/*z* 285.0398 and 593.1513; the loss of 146 Da (739.2091 − 593.1513) and 308 Da (593.1513 − 285.0398) could correspond to kaempferol-*O*-rutinosyl rhamnoside. Likewise, compounds 123, 124, 129, 130, 131, and 132 were tentatively identified as kaempferol coumaroyl rutinoside.

Compounds 72 and 81 were eluted at 7.80 and 8.46 min, with the quasi-molecular ion [M − H]^−^ at *m*/*z* 917.2357 (C_42_H_46_O_23_) and fragment ions [M − H]^−^ at *m*/*z* 301.0343, 463.0869, 609.1456, and 755.2032, indicating that they possessed a quercetin moiety. Considering the neutral loss of 308 Da (917.2357 − 609.1456) and 146 Da (609.1456 − 463.0869 or 917.2357 − 755.2032), they could be quercetin caffeoyl rutinosyl rhamnoside. Likewise, the quasi-molecular ion [M − H]^−^ at *m*/*z* 901.2408 (C_42_H_46_O_22_) of compounds 94, 96, 109, 120, and 125, which indicated a loss of 146 Da (901.2408 − 755.2032), could be quercetin coumaroyl rutinosyl rhamnoside.

Compound 74 appeared at 7.99 min, with the same quasi-molecular ion [M − H]^−^ at *m*/*z* 593.1512 (C_27_H_30_O_15_) and the fragment ion [M − H]^−^ at *m*/*z* 285.0394 as compound 80. Therefore, it was identified as a kaempferol-3-*O*-rutinoside isomer.

Compounds 83 and 85 were obtained at 8.49 and 8.67 min, respectively, with the quasi-molecular ion [M − H]^−^ at *m*/*z* 623.1618 (C_28_H_32_O_16_) and the fragment ion [M − H]^−^ at *m*/*z* 315.0502, indicating the loss of 308 Da; hence, they were identified as isorhamnetin-3-*O*-rutinoside by referring to the literature [30].

Compound 76 possessed a molecular ion [M − H]^−^ at *m*/*z* 433.0776 (C_20_H_18_O_11_) and a fragment ion at *m*/*z* 301.0351 [M − 132.042]^−^, corresponding to the loss of an arabinose moiety; thus, compound 76 was identified as quercetin-*O*-arabinoside.

Compounds 77 and 88 possessed a similar molecular ion [M − H]^−^ at *m*/*z* 447.0933 (C_21_H_20_O_11_) and a fragment ion at *m*/*z* 285.0387 [M − 162.052]^−^, indicating the loss of one glucose moiety. According to [31], compounds 77 and 88 were identified as astragalin isomers.

Compounds 92 and 100 possessed a molecular ion [M − H]^−^ at *m*/*z* 447.0933 (C_21_H_20_O_11_) and a fragment ion at *m*/*z* 301.0341 [M − 146.057]^−^, corresponding to the loss of a rhamnose moiety; therefore, compounds 92 and 100 were identified as quercetin-*O*-rhamnoside.

Compound 133 possessed a molecular ion at *m*/*z* 269.0455 (C_21_H_20_O_11_) and fragment ions at *m*/*z* 117.0334 and 151.0029. Therefore, compound 133 was identified as apigenin.

Compound 134 eluted at 14.87 min, and it possessed a similar molecular ion [M − H]^−^ at *m*/*z* 285.0405 (C_15_H_10_O_6_) and fragment ion at *m*/*z* 151.0021 to kaempferol. Accordingly, compound 134 was identified as a kaempferol isomer.

##### Identification of Flavones

Compound 121 possessed a molecular ion [M − H]^−^ at *m*/*z* 283.0612 (C_16_H_12_O_5_) and fragment ions [M − H]^−^ at *m*/*z* 268.0374, 269.0415, 151.0029, and 107.0128; thus, compound 121 was identified as genkwanin.

Compound 128 possessed a molecular ion [M − H]^−^ at *m*/*z* 431.0984 (C_21_H_20_O_10_) and a fragment ion at *m*/*z* 269.0457. Therefore, compound 128 was identified as oroxin A.

##### Identification of Flavanones

Compounds 37 and 46 possessed a molecular ion [M − H]^−^ at *m*/*z* 449.1089 (C_21_H_22_O_11_) and a fragment ion at *m*/*z* 287.0560 [M − 162.052]^−^, indicating the loss of a hexose moiety. According to [32], compounds 37 and 46 were identified as erodcyol-*O*-hexoside.

##### Identification of Flavanonols

Compounds 19 and 51 eluted at 4.48 and 6.32 min, respectively, with the quasi-molecular ion [M − H]^−^ at *m*/*z* 465.1038 (C_21_H_22_O_12_) and the fragmentation ion [M − H]^−^ at *m*/*z* 303.0507, resulting from the neutral loss of 162 Da (465.1038 − 303.0507) and corresponding to the loss of one hexose moiety. Therefore, compounds 19 and 51 were determined to be taxifolin galactoside and taxifolin glucoside, respectively.

Compound 66 possessed a molecular ion [M − H]^−^ at *m*/*z* 303.0510 (C_15_H_12_O_7_) and fragment ions [M − H]^−^ at *m*/*z* 125.0233, 153.0186, and 151.0026. Therefore, compound 66 was identified as taxifolin.

### 2.4. Pharmacological Activity of Constituents in CT

A total of 135 chemical constituents were identified in CT for the first time, including 67 chlorogenic acid derivatives, 48 flavonoids, and 20 anthocyanins. According to the literature, chlorogenic acid derivatives, including chlorogenic acid, isochlorogenic acid A, and isochlorogenic acid B, exhibit potent anti-inflammatory, antibacterial, antioxidant, analgesic, and antipyretic activities in vitro and in vivo (animal models) [33,34,35,36,37,38]. Quercetin has been reported to have anti-inflammatory, antioxidant, antibacterial, antitumor, and cardiovascular protective effects [39]. Kaempferol has antiosteoporosis and protective effects on damaged tissues, in addition to the above effects [40]. Procyanidins B1 and B2 have been reported to have anti-inflammatory, antibacterial, antioxidant, and other effects [41]. These compounds might be the effective constituents of CT, contributing to its pharmacological activity.

## 3. Materials and Methods

### 3.1. Chemicals and Reference Standards

MS-grade formic acid was purchased from Thermo Fisher Scientific (Carlsbad, CA, USA). Methanol and acetonitrile were of chromatographic grade, provided by Merck (Branchburg, NJ, USA). Water used as the mobile phase solvent was obtained from Watson Water (Guangzhou, China), and the ethanol used in the study was of analytical grade. The reference standards of procyanidin B1 (batch no. wkq19062802) and procyanidin B2 (batch no. wkq19042903) were obtained from Weikeqi Biological Technology Co., Ltd. (Chengdu, Sichuan, China). Reference standards of *trans*-3-caffeoylquinic acid (*trans*-3-CQA, neochlorogenic acid, X-014-170309), *trans*-4-caffeoylquinic acid (*trans*-4-CQA, cryptochlorogenic acid, Y-067-180320), *trans*-5-caffeoylquinic acid (*trans*-5-CQA, chlorogenic acid, L-007-171216), 3,5-dicaffeoylquinic acid (3,5-DiCQA, isochlorogenic acid A, Y-068-170903), 3,4-dicaffeoylquinic acid (3,4-DiCQA, isochlorogenic acid B, Y-069-180105), 4,5-dicaffeoylquinic acid (4,5-DiCQA, isochlorogenic acid C, Y-070-170515), isoquercitrin (Y-076-18106), kaempferol (S-014-171216), and naringenin (Y-030-190812) were provided by Chengdu Herbpurify Co., Ltd. (Chengdu, China). Reference standards of quercetin (AF8041802) and rutin (AF8032520) were provided by Chengdu Alfa Biotechnology Co., Ltd. (Chengdu, China). The reference standard of eriodictyol (PS1160-0025) was provided by Chengdu Push Bio-Technology Co., Ltd. (Chengdu, China). The reference standard of nicotiflorin (CFN99830) was provided by Wuhan Tianzhi Biotechnology Co., Ltd. (Wuhan, Hubei, China). The fresh leaves of CT were obtained from Leye County, Baise city, Guangxi province, and they were dried under vacuum conditions at 45 °C. The specimen (20201013) was stored at the School of Pharmaceutical Sciences, Hunan University of Medicine, Changsha, China.

### 3.2. Reference Standards and Sample Preparation

The dried powder of CT (5 g) was extracted under reflux in 100 mL of 70% aqueous ethanol for 1 h, and then the extracted solution was filtrated and dried under reduced pressure to yield a brown residue, which was dissolved in methanol. The sample was centrifuged at 12,000 rpm for 20 min. A volume of 2 µL was injected into an UHPLC-Q-exactive orbitrap MS for analysis. All reference standards were accurately weighed and dissolved in methanol before storing in a refrigerator at 4 °C until further analysis.

### 3.3. Instruments and Conditions

The instruments used for this study included a Thermo Q-exactive focus orbitrap MS connected to a Thermo Scientific Dionex Ultimate 3000 RS (Thermo Fisher Scientific, Carlsbad, CA, USA). Separation was performed on a Thermo Scientific Hypersil GOLD^TM^ aQ (100 mm × 2.1 mm, 1.9 μm). The column temperature was kept at 35 °C, and the sample was maintained at 10 °C. The mobile phase was water with 0.1% formic acid (A) and acetonitrile (B). The gradient program was as follows: 0 min, 5% B; 2 min, 10% B; 5 min, 20% B; 10 min, 25% B; 12 min, 55% B; 20 min, 80% B; 25 min, 95% B; 26 min, 5% B; and 30 min, 5% B. MS analysis was performed in both positive and negative ionization modes using electrospray ionization (ESI) in the scan range of *m*/*z* 120–1000 at a resolution of 35,000. The source conditions were as follows: sheath gas, 30; auxiliary gas, 10; spray voltage, 3.0 kV for (−)-ESI and 3.5 kV for (+)-ESI; capillary temperature, 320 °C; auxiliary gas heater temperature, 350 °C. The MS^1^ spectra were acquired in full MS mode at a resolution of 35,000, whereas MS^2^ spectra were obtained by ddMS^2^ or parallel reaction monitoring (PRM) mode triggered by inclusion ions [12]. The NEC (normalized collision energy) was set as 30%, with 5.0 × e^5^ of the automatic gain control (AGC) target. Data were processed using Xcalibur™ version 4.1 and Compound Discoverer 3.0 (Thermo Fisher Scientific, Carlsbad, CA, USA).

### 3.4. Prediction of Expected Compounds

It is widely known that chemical constituents in the same category possess an identical carbon skeleton and homologous biosynthetic pathways. CGA analogues constitute a large family of esters formed between quinic acid or shikimic acid and one to four special residues, most commonly *p*-coumaric acid, caffeic acid, sinapic acid, and ferulic acid. Therefore, the molecular structure of the CGA derivatives can be predicted [11,26]. Likewise, flavonoids and anthocyanins can also be predicted. Quercetin and kaempferol are the carbon skeletons of flavonoids connected by hydroxyl (OH) and glycoside bonds; their structures differ in terms of the type and number of sugar units, e.g., glucose (C_6_H_10_O_5_), arabinose (C_5_H_8_O_4_), rhamnose (C_6_H_10_O_4_), rutinose (C_12_H_20_O_9_), and glucosyl rutinose (C_18_H_30_O_14_).

### 3.5. Establishment of Diagnostic Fragmentation Ions (DFIs) and Neutral Loss (NL)

CGAs, anthocyanidins, and flavonoids with the same carbon skeleton were expected to have similar fragment ions. The fragment ion patterns of six CGAs, six flavonoids, and two anthocyanins were investigated using UHPLC-Q-exactive orbitrap MS in negative mode. The fragmentation pathway of CGAs is shown in Figure 2A. The common fragmentation ions were identified as 191.056 (C_7_H_11_O_6_), 173.045 (C_7_H_10_O_5_), 179.034 (C_9_H_7_O_3_), and 135.045 (C_8_H_7_O_2_), which could be considered diagnostic fragmentation ions. The neutral losses, including C_9_H_6_O_3_, C_7_H_10_O_5_, H_2_O, and CO_2_, are summarized in Figure 2A. Likewise, the diagnostic fragmentation ions (151.002, C_7_H_3_O_4_; 107.012, C_6_H_3_O_2_) and neutral losses (C_7_H_4_O_4_ and C_6_H_10_O_5_) of flavonoids are displayed in Figure 2B. The diagnostic fragmentation ions of 289.072 (C_15_H_13_O_6_), 407.077 (C_22_H_15_O_8_), 245.081 (C_14_H_13_O_4_), 125.023 (C_6_H_5_O_3_), and 161.023 (C_9_H_5_O_3_), along with neutral losses (C_15_H_12_O_6_, C_8_H_10_O_4_, C_6_H_8_O_3_, CO_2_, and C_9_H_8_O_3_), are shown in Figure 2C.

## 4. Conclusions

In this study, a rapid and effective method for identifying the chemical constituents of CT was developed using UHPLC-Q-exactive orbitrap combined with PRM; the compounds were predicted using DFI and NL techniques. A total of 135 compounds were identified, comprising 67 chlorogenic acid derivatives, 48 flavonoids, and 20 anthocyanins, all of which are reported for the first time in CT. These results expand the knowledge on the chemical composition of CT and provide a scientific basis for the subsequent elucidation of the medicinal substances present and their activities, enabling further development and utilization of this plant. Overall, the results lay the foundation for in-depth research on the pharmacodynamic basis of CT. Furthermore, this research strategy can be used for the characterization of various samples.

## Figures and Tables

**Figure 1 molecules-27-04038-f001:**
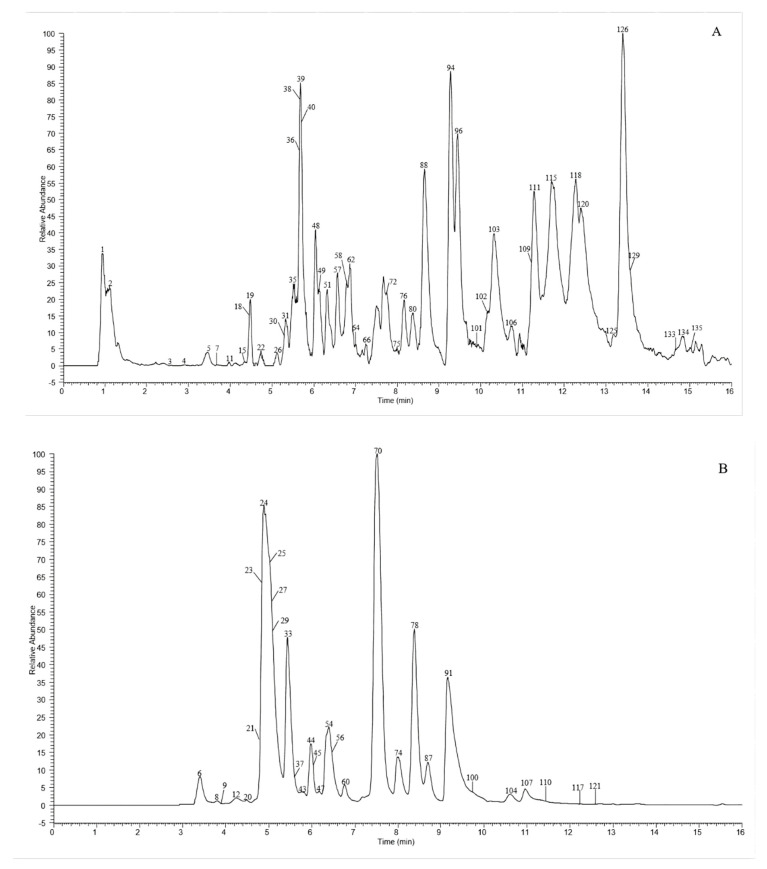
High-resolution extracted ion chromatogram (HREIC) in 5 ppm for multiple compounds in CT: (**A**) *m*/*z* 269.04554, 271.06119, 283.06119, 285.04046, 287.05611, 289.07176, 301.03537, 303.05102, 353.10893, 431.09837, 433.07763, 465.10384, 497.10893, 497.13006, 529.13514, 579.1497, 625.14102, 677.15119, and 901.24079; (**B**) *m*/*z* 335.07724, 337.09289, 341.08780, 353.08780, 367.10345, 397.11402, 447.09328, 449.10893, 497.13006, 499.12458, 499.14571, 515.11949, 515.14062, 529.15627, 593.15119, 609.1461, 623.16175, 677.19345, 755.20401, 771.19893, and 917.23571; (**C**) *m*/*z* 335.07724, 337.09289, 367.10345, 397.11402, 449.10893, 497.13006, 499.12458, 499.14571, 515.14062, 529.15627, 623.16175, 677.19345, 677.15119, 771.19893, and 917.23571; (**D**) *m*/*z* 757.21856, 597.14501, 741.20252, 465.10275, 595.16574, 625.17631, 479.11840, 609.18139, 579.17083, 741.22365, and 757.19744; (**E**) *m*/*z* 291.08631, 463.12348, 479.11840, 579.1497, 611.16066, and 741.22365. (**A**–**C**) EIC in negative mode; (**D**,**E**) EIC in positive mode.

**Figure 2 molecules-27-04038-f002:**
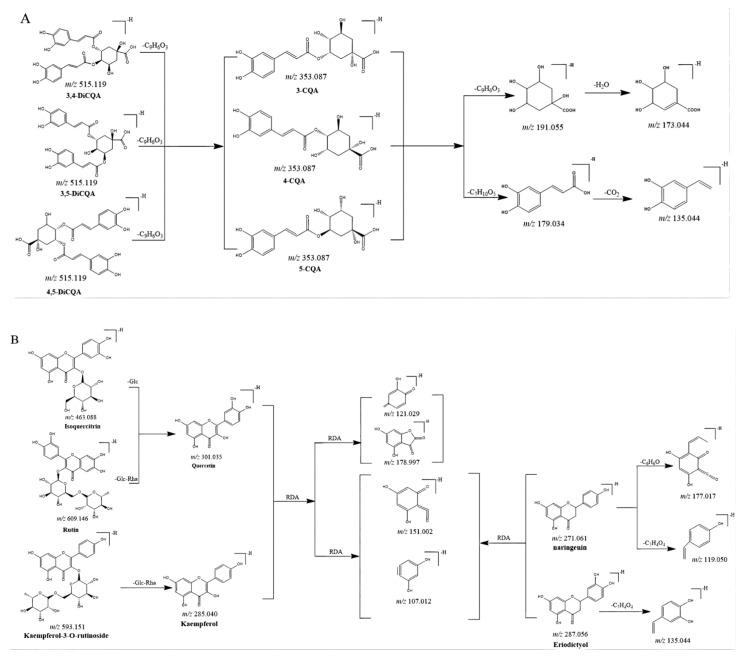
Proposed selected fragmentation patterns of components identified in CT: CGAs (**A**); flavonoids (**B**); B-type procyanidin (**C**).

**Table 1 molecules-27-04038-t001:** Retention times and mass spectral data of *Cephalanthus tetrandrus* (Roxb.) Ridsd. et Badh. F.

Peak	t_R_	Theoretical Mass *m*/*z*	Experimental Mass *m*/*z*	Error (ppm)	Formula	MS/MS Fragment (−)	MS/MS Fragment (+)	Identification
1	0.93	353.1089	353.1082	−2.14	C_13_H_22_O_11_	MS^2^ [353]: 353.1088 (100), 173.0444 (67), 191.0551 (62), 85.0279 (3)		QA-hexoside
2	1.11	353.1089	353.1084	−1.51	C_13_H_22_O_11_	MS^2^ [353]: 191.0551 (100), 173.0443 (21), 179.0549 (15), 129.0376 (11)		QA-hexoside
3	2.58	341.0878	341.0876	−0.63	C_15_H_18_O_9_	MS^2^ [341]: 161.0233 (100), 179.0340 (41), 135.0439 (20)		CA-hexoside
4	2.96	341.0878	341.0875	−0.84	C_15_H_18_O_9_	MS^2^ [341]: 161.0232 (100), 179.0338 (55), 135.0438 (29)		CA-hexoside
5	3.38	341.0878	341.0872	−1.72	C_15_H_18_O_9_	MS^2^ [341]: 203.0341 (100), 161.0233 (93), 135.0438 (53), 179.0342 (49), 101.0230 (25)		CA-hexoside
6 *	3.41	353.0878	353.0874	−1.15	C_16_H_18_O_9_	MS^2^ [353]: 191.0552 (100), 135.0440 (79), 179.0341 (46), 173.0448 (2)		*trans*-3-CQA
7	3.66	341.0878	341.0874	−1.19	C_15_H_18_O_9_	MS^2^ [341]: 161.0233 (100), 179.0339 (91), 135.0438 (26)		CA-hexoside
8	3.80	341.0878	341.0873	−1.36	C_15_H_18_O_9_	MS^2^ [341]: 161.0233 (100), 179.0339 (23)		CA-hexoside
9	3.81	515.1406	515.1385	−4.15	C_22_H_28_O_14_	MS^2^ [515]: 191.0554 (100)		CQA-hexoside
10	3.82	499.1457	499.1463	1.11	C_22_H_28_O_13_	MS^2^ [499]: 173.0446 (100), 93.0332 (89), 191.0553 (38), 163.0392 (22)		4-pCoQA-hexoside
11	4.01	341.0878	341.0875	−0.92	C_15_H_18_O_9_	MS^2^ [341]: 179.0340 (100), 135.0439 (26), 161.0232 (17)		CA-hexoside
12	4.21	341.0878	341.0876	−0.75	C_15_H_18_O_9_	MS^2^ [341]: 179.0340 (100), 161.0233 (37), 135.0439 (27)		CA-hexoside
13 *	4.25	577.1351	577.1350	−0.35	C_30_H_26_O_12_	MS^2^ [577]: 289.0716 (100), 125.0232 (88), 407.0769 (55), 161.0235 (25), 245.0813 (15), 353.0880 (19)	MS^2^ [579]: 127.0390 (100), 139.0391 (30), 123.0443 (21)	Procyanidin B1
		579.1497	579.1517	3.50				
14	4.29	337.0929	337.0926	−0.89	C_16_H_18_O_8_	MS^2^ [337]: 163.0389 (100),		*trans*-3-pCoQA
15	4.31	515.1406	515.1408	0.35	C_22_H_28_O_14_	MS^2^ [515]: 191.0554 (100), 323.0773 (39), 161.0235 (11)		CQA-hexoside
16	4.40	529.1563	529.1566	0.65	C_23_H_30_O_14_	MS^2^ [529]: 191.0554 (100), 173.0446 (41)		5-FQA-hexoside
17	4.45	677.1935	677.1924	−1.49	C_28_H_38_O_19_	MS^2^ [677]: 191.0553 (100), 353.0882 (22)		CQA-Dihexoside
18	4.47	341.0878	341.0875	−0.84	C_15_H_18_O_9_	MS^2^ [341]: 179.0340 (100), 161.0233 (39), 135.0439 (29)		CA-hexoside
19	4.48	465.1038	465.1030	−1.76	C_21_H_22_O_12_	MS^2^ [465]: 61.9869 (100), 285.0400 (88), 125.0231 (53), 275.0556 (35), 177.0183 (26), 178.9975 (21), 303.0507 (18),151.0023 (11)		Taxifolin hexoside
20	4.50	515.1406	515.1397	−1.78	C_22_H_28_O_14_	MS^2^ [515]: 191.0554 (100), 323.0773 (39), 161.0235 (12)		CQA-hexoside
21	4.69	341.0878	341.0877	−0.37	C_15_H_18_O_9_	MS^2^ [341]: 179.0340 (100), 161.0233 (30), 135.0439 (24)		CA-hexoside
22	4.74	515.1406	515.1395	−2.25	C_22_H_28_O_14_	MS^2^ [515]: 191.0554 (100), 179.0342 (21), 323.0774 (18), 173.0447 (15)		CQA-hexoside
23	4.78	499.1457	499.1449	−1.61	C_22_H_28_O_13_	MS^2^ [499]: 191.0554 (100), 173.0446 (36), 163.0391 (21)		5-pCoQA-hexoside
24 *	4.89	353.0878	353.0872	−1.74	C_16_H_18_O_9_	MS^2^ [353]: 173.0448 (100), 179.0342 (69), 191.0554 (58), 135.0441 (29)		*trans*-4-CQA
25 *	5.03	353.0878	353.0872	−1.66	C_16_H_18_O_9_	MS^2^ [353]: 191.0554 (100)		*trans*-5-CQA
26	5.14	757.2186	757.2186	0.08	C_33_H_41_O_20_		MS^2^ [757]: 287.0534 (100), 449.1057 (57), 595.1647 (1)	Cyanidin-*O*-rutinoside-*O*-galactoside
27	5.21	397.1140	397.1133	−1.71	C_18_H_22_O_10_	MS^2^ [397]: 191.0553 (100), 173.0446 (12)		*trans*-3-SQA
28	5.25	337.0929	337.0925	−1.07	C_16_H_18_O_8_	MS^2^ [337]: 173.0444 (100), 163.0389 (20), 191.0551 (12)		*trans*-4-pCoQA
29	5.25	529.1563	529.1561	−0.28	C_23_H_30_O_14_	MS^2^ [529]: 191.0554 (100), 173.0447 (31), 193.0499 (15)		5-FQA-hexoside
30	5.30	397.1140	397.1131	−2.24	C_18_H_22_O_10_	MS^2^ [397]: 191.0553 (100), 173.0446 (8), 179.0340 (6)		*cis*-3-SQA
31	5.31	515.1406	515.1397	−1.78	C_22_H_28_O_14_	MS^2^ [515]: 191.0554 (100), 323.0773 (38), 161.0235 (14)		CQA-hexoside
32 *	5.34	577.1351	577.1346	−0.88	C_30_H_26_O_12_	MS^2^ [577]: 289.0720 (100), 125.0233 (92), 407.0775 (60), 161.0236 (24), 245.0818 (16)	MS^2^ [579]: 127.0389 (100), 139.0388 (53), 123.0441 (26)	Procyanidin B2
		579.1497	579.1484	−2.28				
33	5.44	353.0878	353.0874	−1.23	C_16_H_18_O_9_	MS^2^ [353]: 191.0551 (100)		*cis*-5-CQA
34	5.52	515.1406	515.1396	−2.02	C_22_H_28_O_14_	MS^2^ [515]: 191.0554 (100), 353.1092 (16)		CQA-hexoside
35	5.54	497.1301	497.1289	−2.26	C_22_H_26_O_13_	MS^2^ [497]: 179.0341 (100), 335.0775 (24), 135.0441 (20) 161.0235 (18)		CSA-hexoside
36	5.54	771.1989	771.1998	1.09	C_33_H_40_O_21_	MS^2^ [771]: 301.0349 (100), 300.0266 (56),463.0845 (15), 609.1459 (7)		Quercetin-*O*-glucosylrutinoside
37	5.62	449.1089	449.1088	−0.39	C_21_H_22_O_11_	MS^2^ [449]: 61.9869 (100), 259.0607 (30), 269.0451 (21), 125.0230 (12), 287.0560 (11)		Erodcyol-*O*-hexoside
38	5.67	335.0772	335.0772	−0.00	C_16_H_16_O_8_	MS^2^ [335]: 161.0232 (100), 179.0339 (25), 135.0439 (15)		3-CQL
39	5.68	289.0718	289.0716	−0.70	C_15_H_14_O_6_	MS^2^ [289]: 245.0815 (100), 109.0282 (69), 125.0231 (66), 203.0704 (64)	MS^2^ [291]: 139.0388 (100), 123.0440 (48), 147.0439 (20)	Catechin
		291.0863	291.0859	−1.39				
40	5.74	529.1563	529.1563	0.06	C_23_H_30_O_14_	MS^2^ [529]: 191.0553 (100), 193.0498 (43), 173.0445 (14)		5-FQA-hexoside
41	5.76	497.1301	497.1303	0.56	C_22_H_26_O_13_	MS^2^ [497]: 161.0234 (100), 173.0809 (49), 193.0709 (41)		CQL-hexoside
42	5.79	337.0929	337.0928	−0.36	C_16_H_18_O_8_	MS^2^ [337]: 173.0444 (100), 163.0390 (21), 119.0498 (5),		*cis*-4-pCoQA
43	5.85	335.0772	335.0771	−0.45	C_16_H_16_O_8_	MS^2^ [335]: 179.0339 (100), 135.0439 (39), 161.0233 (16)		5-CSA
44	5.98	677.1935	677.1951	2.39	C_28_H_38_O_19_	MS^2^ [677]: 191.0553 (100)		CQA-dihexoside
45	5.99	353.0878	353.0874	−1.06	C_16_H_18_O_9_	MS^2^ [353]: 191.0551 (100), 173.0445 (2), 85.0280 (1)		*cis*-3-CQA
46	5.99	449.1089	449.1080	−2.01	C_21_H_22_O_11_	MS^2^ [449]: 61.9869 (100), 269.0450 (16), 259.0593 (14), 287.0560 (6)		Erodcyol-*O*-hexoside
47	6.01	529.1563	529.1553	−1.91	C_23_H_30_O_14_	MS^2^ [529]: 173.0447 (100), 191.0554 (74), 193.0499 (58)		4-FQA-hexoside
48	6.02	757.2186	757.2177	−1.22	C_33_H_41_O_20_		MS^2^ [757]: 287.0533 (100), 449.1054 (59), 595.1649 (1)	Cyanidin-*O*-rutinoside-*O*-galactoside
49	6.17	335.0772	335.0769	−1.02	C_16_H_16_O_8_	MS^2^ [335]: 179.0340 (100), 135.0439 (44), 161.0233 (20)		4-CSA
50	6.25	367.1035	367.1028	−1.68	C_17_H_20_O_9_	MS^2^ [367]: 173.0445 (100), 191.0553 (68), 193.0497 (23)		4-FQA
51	6.32	465.1038	465.1038	−0.19	C_21_H_22_O_12_	MS^2^ [465]: 285.0403 (100), 125.0231 (54), 177.0182 (29), 303.0508 (19), 139.0387 (12)		Taxifolin hexoside
52	6.34	337.0929	337.0927	−0.71	C_16_H_18_O_8_	MS^2^ [337]: 191.0551 (100),173.0444 (2)		*cis*-5-pCoQA
53	6.37	335.0772	335.0769	−0.81	C_16_H_16_O_8_	MS^2^ [335]: 161.0235 (100), 135.0440 (51), 179.0341 (10), 173.0446 (2)		1-CQL
54	6.39	397.1140	397.1141	−0.38	C_18_H_22_O_10_	MS^2^ [397]: 161.0235 (100), 191.0554 (26), 173.0447 (13)		*trans*-5-SQA
55	6.42	367.1035	367.1028	−1.76	C_17_H_20_O_9_	MS^2^ [367]: 191.0551 (100), 93.0331 (23), 173.0444 (14)		*trans*-5-FQA
56	6.49	335.0772	335.0772	−0.27	C_16_H_16_O_8_	MS^2^ [335]: 161.0232 (100), 135.0438 (12), 179.0339 (10)		4-CQL
57	6.57	397.1140	397.1136	−0.93	C_18_H_22_O_10_	MS^2^ [397]: 191.0553 (100), 173.0446 (24)		*cis*-5-SQA
58	6.63	771.1989	771.1989	0.05	C_33_H_40_O_21_	MS^2^ [771]: 367.1034 (100), 609.1469 (31), 301.0352 (17), 463.0880 (6)		Quercetin-*O*-glucosylrutinoside
59	6.74	367.1035	367.1029	−1.32	C_17_H_20_O_9_	MS^2^ [367]: 191.0551 (100)		*cis*-5-FQA
60	6.77	755.2040	755.2032	−1.02	C_33_H_40_O_20_	MS^2^ [755]: 300.0271 (100), 191.0551 (37), 301.0391 (14)		Quercetin rhamnosylrutinoside
61	6.83	515.1195	515.1181	−2.60	C_25_H_24_O_12_	MS^2^ [515]: 191.0552 (100), 323.0558 (16), 173.0444 (14), 179.0338 (3)		Dicaffeoylquinic acid
62	6.86	625.1410	625.1392	−2.79	C_27_H_30_O_17_	MS^2^ [625]: 301.0348 (100), 300.0271 (94), 625.1396 (66), 463.0883 (3), 151.0023 (2)		Quercetin-*O*-sophoroside
63	6.89	771.1989	771.1961	−3.66	C_33_H_40_O_21_	MS^2^ [771]: 301.0350 (100), 300.0271 (33), 463.0898 (6), 609.1464 (4)		Quercetin-*O*-glucosylrutinoside
64	7.11	597.1450	597.1441	−1.53	C_26_H_29_O_16_		MS^2^ [597]: 303.0493 (100)	Delphinidin-3-xylosylglucoside
65	7.24	739.2091	739.2125	4.62	C_33_H_40_O_19_	MS^2^ [739]: 284.0322 (100), 285.0398 (19), 593.1513 (1)	MS^2^ [741]: 287.0544 (100)	Kaempferol-*O*-rutinosylrhamnoside
		741.2237	741.2226	−1.34				
66	7.26	303.0510	303.0504	−2.00	C_15_H_12_O_7_	MS^2^ [303]: 125.0233 (100), 153.0186 (12), 175.0393 (11), 151.0026 (10), 199.0398 (10)		Taxifolin
67	7.50	449.1078	449.1073	−1.00	C_21_H_21_O_11_		MS^2^ [449]: 303.0495 (100)	Delphinidin-3-*O*-rhamnoside
68	7.50	611.1607	611.1597	−1.43	C_27_H_31_O_16_		MS^2^ [611]: 303.0493 (100)	Delphinidin-3-*O*-rutinoside
69	7.51	465.1028	465.1019	−1.64	C_21_H_21_O_12_		MS^2^ [465]: 303.0494 (100)	Delphinidin-3-*O*-hexoside
70 *	7.52	609.1461	609.1455	−0.88	C_27_H_30_O_16_	MS^2^ [609]: 300.0271 (100), 301.0345 (57)		Rutin
71 *	7.77	463.0882	463.0877	−0.93	C_21_H_20_O_12_	MS^2^ [463]: 300.0272 (100), 301.0344 (56), 151.0024 (4), 178.9977 (3)	MS^2^ [465]: 303.0496 (100)	Isoquercitrin
		465.1028	465.1022	−1.12				
72	7.80	917.2357	917.2340	−1.79	C_42_H_46_O_23_	MS^2^ [917]: 301.0343 (100), 300.0271 (95), 609.1456 (15), 463.0869 (12), 151.0027 (8), 771.1989 (1)		Quercetin caffeoylrutinosylrhamnoside
73	7.98	595.1657	595.1653	−0.66	C_27_H_31_O_15_		MS^2^ [595]: 287.0544 (100)	Cyanidin-3-*O*-rutinoside
74	7.99	593.1512	593.1508	−0.53	C_27_H_30_O_15_	MS^2^ [593]: 284.0322 (100), 285.0394 (53), 151.0026 (3)		Kaempferol-3-*O*-rutinoside isomer
75 *	8.06	515.1195	515.1190	−0.81	C_25_H_24_O_12_	MS^2^ [515]: 173.0444 (100), 179.0339 (92), 191.0551 (38), 135.0439 (14), 161.0231 (13), 353.0877 (12)		3,4-DiCQA
76	8.15	433.0776	433.0759	−3.87	C_20_H_18_O_11_	MS^2^ [433]: 300.0278 (100), 123.0440 (24), 119.0339 (12), 301.0351 (12), 151.0026 (1)		Quercetin-*O*-arabinoside
77	8.34	447.0933	447.0928	−0.95	C_21_H_20_O_11_	MS^2^ [447]: 284.0323 (100), 285.0387 (27), 255.0291 (11), 151.0027 (4)	MS^2^ [449]: 287.0544 (100)	Astragalin isomer
		449.1078	449.1071	−1.60				
78 *	8.34	515.1195	515.1187	−1.42	C_25_H_24_O_12_	MS^2^ [515]: 191.0551 (100), 179.0339 (80), 173.0444 (37), 353.0874 (15), 135.0438 (14)		3,5-DiCQA
79	8.38	595.1657	595.1647	−1.61	C_27_H_31_O_15_		MS^2^ [595]: 287.0544 (100)	Cyanidin-3-*O*-rutinoside isomer
80 *	8.39	593.1512	593.1506	−0.95	C_27_H_30_O_15_	MS^2^ [593]: 285.0400 (100), 284.0323 (49), 151.0026 (1)		Kaempferol-3-*O*-rutinoside
81	8.46	917.2357	917.2359	0.27	C_42_H_46_O_23_	MS^2^ [917]: 300.0268 (100), 301.0345 (52), 609.1443 (12), 151.0025 (4), 463.0871 (1)		Quercetin caffeoylrutinosylrhamnoside
82	8.48	625.1763	625.1754	−1.43	C_28_H_33_O_16_		MS^2^ [625]: 317.0649 (100)	Petunidin-3-*O*-rutinoside
83	8.49	623.1618	623.1608	−1.39	C_28_H_32_O_16_	MS^2^ [623]: 314.0428 (100), 315.0502 (77), 299.0185 (13), 101.0230 (10)		Isorhamnetin-3-*O*-rutinoside
84	8.67	479.1184	479.1178	−1.26	C_22_H_23_O_12_		MS^2^ [479]: 317.0650 (100), 163.0387 (25)	Petunidin hexoside
85	8.67	623.1618	623.1613	−0.70	C_28_H_32_O_16_	MS^2^ [623]: 315.0505 (100), 314.0428 (39)		Isorhamnetin-3-*O*-rutinoside
86	8.67	625.1763	625.1757	−0.95	C_28_H_33_O_16_		MS^2^ [625]: 317.0650 (100)	Petunidin-3-*O*-rutinoside isomer
87	8.68	289.0718	289.0714	−1.11	C_15_H_14_O_6_	MS^2^ [289]: 109.0283 (100), 123.0441 (77), 125.0234 (57), 151.0391 (27), 137.0234 (27), 203.0710 (20)	MS^2^ [291]: 139.0387 (100), 123.0440 (49), 147.0438 (20)	Epicatechin
		291.0868	291.0858	−1.84				
88	8.73	447.0933	447.0929	−0.79	C_21_H_20_O_11_	MS^2^ [447]: 284.0323 (100), 285.0387 (42), 255.0298 (10), 151.0022 (2)	MS^2^ [449]: 287.0545 (100)	Astragalin isomer
		449.1078						
89	8.77	479.1184	479.1180	−0.64	C_22_H_23_O_12_		MS^2^ [479]: 317.0650 (100), 163.0387 (29)	Petunidin hexoside
90	8.99	479.1184	479.1181	−0.70	C_22_H_23_O_12_		MS^2^ [479]: 317.0650 (100)	Petunidin hexoside
91	9.15	515.1195	515.1188	−1.28	C_25_H_24_O_12_	MS^2^ [515]: 173.0445 (100), 179.0340 (74), 191.0552 (24), 353.0874 (16), 135.0438 (10)		1,4-DiCQA
92	9.20	447.0933	447.0929	−0.79	C_21_H_20_O_11_	MS^2^ [447]: 300.0273 (100), 301.0341 (34), 285.0403 (14), 151.0026 (2)		Quercetin-*O*-rhamnoside
93	9.25	609.1814	609.1804	−1.51	C_28_H_33_O_15_		MS^2^ [609]: 301.0690 (100)	Peonidin-3-*O*-rutinoside
94	9.27	901.2408	901.2406	−0.22	C_42_H_46_O_22_	MS^2^ [901]: 901.2413 (100), 300.0279 (63), 755.2086 (36), 301.0333 (7)	MS^2^ [903]: 303.0494 (100), 147.0438 (79)	Quercetin coumaroylrutinosylrhamnoside
		903.2553	903.2533	−2.27				
95	9.43	463.1235	463.1236	0.44	C_22_H_23_O_11_		MS^2^ [463]: 301.0689 (100), 121.0279 (94), 147.0432 (31), 139.0382 (25)	Peonidin-*O*-hexoside
96	9.44	901.2408	901.2410	0.33	C_42_H_46_O_22_	MS^2^ [901]: 901.2418 (100), 300.0272 (58), 755.2048 (30), 301.0338 (1),151.0025 (1)	MS^2^ [903]: 303.0492 (100), 147.0437 (82)	Quercetin coumaroylrutinosylrhamnoside
		903.2553	903.2538	−1.66				
97	9.46	499.1246	499.1227	−3.64	C_25_H_24_O_11_	MS^2^ [499]: 173.0444 (100), 163.0389 (21), 179.0337 (7), 191.0557 (4), 135.0436 (1)		4-pCo,5CQA
98	9.54	579.1708	579.1724	2.87	C_27_H_31_O_14_		MS^2^ [579]: 271.0602 (100), 207.0653 (59), 163.0391 (57)	Pelargonidin-3-*O*-rutinoside
99	9.76	499.1246	499.1241	−0.81	C_25_H_24_O_11_	MS^2^ [499]: 191.0551 (100), 179.0341 (45), 173.0447 (12), 135.0440 (10)		3C,5-pCoQA
100	9.77	447.0933	447.0928	−0.95	C_21_H_20_O_11_	MS^2^ [447]: 300.0271 (100), 301.0341 (37), 169.0491 (24), 151.0025 (3)		Quercetin-*O*-rhamnoside
101	9.88	529.1351	529.1344	−1.30	C_26_H_26_O_12_	MS^2^ [529]: 191.0551 (100), 179.0339 (42), 163.0389 (30), 173.0445 (18), 135.0439 (10)		3C,5FQA
102	10.18	529.1351	529.1339	−2.23	C_26_H_26_O_12_	MS^2^ [529]: 193.0496 (100), 173.0443 (10)		3F,5CQA
103	10.31	529.1351	529.1348	−0.60	C_26_H_26_O_12_	MS^2^ [529]: 191.0551 (100), 179.0339 (41), 173.0444 (16), 135.0437 (11)		3C,5FQA
104 *	10.60	515.1195	515.1192	−0.47	C_25_H_24_O_12_	MS^2^ [515]: 173.0446 (100), 179.0341 (77), 191.0554 (37), 353.0878 (19)		4,5-DiCQA
105	10.62	917.2357	917.2354	−0.33	C_42_H_46_O_23_	MS^2^ [917]: 300.0272 (100), 301.0333 (13), 755.2038 (5), 151.0029 (1),		Quercetin caffeoylrutinosylrhamnoside
106 *	10.73	287.0561	287.0560	−0.29	C_15_H_12_O_6_	MS^2^ [287]: 135.0441 (100), 107.0126 (22), 151.0028 (18)		Eriodictyol
107	10.96	515.1195	515.1185	−1.77	C_25_H_24_O_12_	MS^2^ [515]: 161.0232 (100)		Dicaffeoylquinic acid
						179.0339 (56), 173.0445 (33), 135.0439 (14), 191.0546 (11), 353.0871 (5)		
108	10.96	755.1829	755.1823	−0.69	C_36_H_36_O_18_	MS^2^ [755]: 609.1457 (100), 301.0349 (55), 300.0275 (30), 151.0026 (1)	MS^2^ [757]: 147.0438 (100), 303.0494 (16)	Quercetin coumaroylrutinoside
		757.1974	757.1955	−2.48				
109	11.20	901.2408	901.2392	−1.70	C_42_H_46_O_22_	MS^2^ [901]: 300.0272 (100), 755.2032 (86), 301.0339 (21), 151.0028 (2)		Quercetin coumaroylrutinosylrhamnoside
110	11.24	497.1089	497.1080	−1.88	C_25_H_22_O_11_	MS^2^ [497]: 179.0341 (100), 135.0440 (43), 161.0233 (25)		DiCSA
111	11.29	529.1351	529.1347	−0.72	C_26_H_26_O_12_	MS^2^ [529]: 173.0444 (100), 179.0339 (41), 191.0551 (22)		4C,5FQA
112	11.34	499.1246	499.1240	−1.01	C_25_H_24_O_11_	MS^2^ [499]: 173.0444 (100), 179.0339 (41), 191.0551 (22)		*trans*-4-pCo,5CQA
113	11.37	755.1829	755.1825	−0.51	C_36_H_36_O_18_	MS^2^ [755]: 609.1461 (100), 301.0349 (66), 300.0270 (36), 151.0027 (1)	MS^2^ [757]: 147.0438 (100), 303.0495 (20)	Quercetin coumaroylrutinoside
		757.1974	757.1964	−1.28				
114	11.69	529.1351	529.1350	−0.26	C_26_H_26_O_12_	MS^2^ [529]: 173.0444 (100), 179.0338 (60), 191.0550 (47), 135.0438 (10)		4F,5CQA
115	11.72	497.1089	497.1087	−0.35	C_25_H_22_O_11_	MS^2^ [497]: 179.0339 (100), 161.0233 (97), 135.0439 (50)		DiCSA
116	12.18	755.1829	755.1821	−0.92	C_36_H_36_O_18_	MS^2^ [755]: 609.1462 (100), 301.0351 (62), 300.0270 (25), 151.0025 (1)	MS^2^ [757]: 147.0432 (100), 303.0494 (18)	Quercetin coumaroylrutinoside
		757.1974	757.1966	−1.03				
117 *	12.24	301.0354	301.0349	−1.35	C_15_H_10_O_7_	MS^2^ [301]: 151.0025 (100), 178.9975 (53), 121.0281 (14), 107.0126 (9)		Quercetin
118	12.29	497.1089	497.1089	0.09	C_25_H_22_O_11_	MS^2^ [497]: 161.0234 (100), 335.0772 (44), 179.0341 (37), 137.0233 (33), 135.0440 (23)		DiCQL
119	12.32	755.1829	755.1821	−1.00	C_36_H_36_O_18_	MS^2^ [755]: 609.1457 (100), 301.0352 (56), 300.0270 (26), 151.0024 (1)	MS^2^ [757]: 147.0432 (100), 303.0496 (14)	Quercetin coumaroylrutinoside
		757.1974	757.1962	−1.52				
120	12.40	901.2408	901.2391	−1.84	C_42_H_46_O_22_	MS^2^ [901]: 901.2434 (100), 300.0272 (39), 755.2037 (38), 301.0364 (8), 151.0025 (2)		Quercetin coumaroylrutinosylrhamnoside
121	12.43	283.0612	283.0607	−1.44	C_16_H_12_O_5_	MS^2^ [283]: 268.0374 (100), 269.0415 (11), 240.0427 (6), 151.0029 (2), 107.0128 (2)		Genkwanin
122	12.67	499.1246	499.1241	−0.87	C_25_H_24_O_11_	MS^2^ [499]: 173.0444 (100), 179.0340 (49), 191.0551 (32)		*cis*-4-pCo,5CQA
123	12.73	739.1879	739.1873	−0.88	C_36_H_36_O_17_	MS^2^ [739]: 285.0402 (100), 593.1505 (45), 739.1868 (22), 284.0320 (10), 145.0283 (5), 151.0026 (1)	MS^2^ [741]: 147.0439 (100), 287.0545 (18)	Kaempferol coumaroylrutinoside
		741.2025	741.2013	−1.59				
124	13.23	739.1879	739.1879	−0.06	C_36_H_36_O_17_	MS^2^ [739]: 285.0402 (100), 593.1508 (35), 284.0323 (28), 145.0283 (34),151.0025 (1)	MS^2^ [741]: 147.0439 (100), 287.0545 (61)	Kaempferol coumaroylrutinoside
		741.2025	741.2017	−1.09				
125	13.39	901.2408	901.2398	−1.03	C_42_H_46_O_22_	MS^2^ [901]: 755.2032 (100), 300.0272 (99), 301.0343 (26), 756.2083 (13), 151.0027 (1)		Quercetin coumaroylrutinosylrhamnoside
126 *	13.41	271.0612	271.0610	−0.47	C_15_H_12_O_5_	MS^2^ [271]: 151.0025 (100), 119.0498 (39), 177.0182 (13), 107.0124 (12)	MS^2^ [273]: 273.0751 (100), 153.0180 (70),	Naringenin
		273.0758	273.0756	−0.22				
127	13.41	677.1512	677.1492	−2.81	C_34_H_30_O_15_	MS^2^ [677]: 353.0879 (100), 173.0446 (28), 179.0340 (24), 191.0553 (18), 335.0776 (15)		TriCQA
128	13.48	431.0984	431.0976	−1.72	C_21_H_20_O_10_	MS^2^ [431]: 269.0457 (100)		Oroxin A
129	13.50	739.1879	739.1878	−0.22	C_36_H_36_O_17_	MS^2^ [739]: 285.0402 (100), 593.1517 (50), 284.0322 (11), 151.0024 (2)	MS^2^ [741]: 147.0438 (100), 287.0544 (10)	Kaempferol coumaroylrutinoside
		741.2025	741.2017	−1.01				
130	13.60	739.1879	739.1878	−0.14	C_36_H_36_O_17_	MS^2^ [739]: 285.0401 (100), 593.1510 (51), 284.0326 (12), 145.0284 (2), 151.0024 (2)		Kaempferol coumaroylrutinoside
131	13.85	739.1879	739.1878	−0.22	C_36_H_36_O_17_	MS^2^ [739]: 285.0401 (100), 593.1508 (43), 284.0324 (23), 151.0025 (1)	MS^2^ [741]: 147.0438 (100), 287.0545 (94)	Kaempferol coumaroylrutinoside
		741.2025	741.2023	−0.26				
132	14.66	739.1879	739.1875	−0.56	C_36_H_36_O_17_	MS^2^ [739]: 285.0402 (100), 593.1509 (13), 284.0326 (39), 151.0027 (1)	MS^2^ [741]: 287.0545 (100), 147.0439 (70)	Kaempferol coumaroylrutinoside
		741.2025	741.2019	−0.84				
133	14.73	269.0455	269.0451	−1.66	C_15_H_10_O_5_	MS^2^ [269]: 117.0334 (100), 151.0029 (33), 107.0127 (33), 121.0284 (10)	MS^2^ [271]: 153.0183 (37), 119.0494 (34), 109.1016 (19)	Apigenin
		271.0601	271.0602	0.44				
134	14.87	285.0405	285.0403	−0.43	C_15_H_10_O_6_	MS^2^ [285]: 285.0406 (100), 185.0600 (23), 107.0125 (18), 137.0235 (15), 143.0493 (14), 159.0444 (12), 151.0021 (1)		Kaempferol isomer
135 *	14.98	285.0405	285.0400	−1.37	C_15_H_10_O_6_	MS^2^ [285]: 285.0397 (100), 178.9978 (5), 151.0023 (2), 107.0121 (1)		Kaempferol

* With standard references.

## Data Availability

Data will be provided upon request.

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
