# Peer review of "Rapid Identification of Constituents in Cephalanthus tetrandrus (Roxb.) Ridsd. et Badh. F. Using UHPLC-Q-Exactive Orbitrap Mass Spectrometry"

_molecules, 2022, doi:10.3390/molecules27134038_

Round 1

Reviewer 1 Report

In the abstract part:

How Chlorogenic acids derivatives are 67????

The conclusion should be at the last (both materials and conclusion are numerated 4). 

Reviewer 2 Report

The manuscript presents interesting and valuable work, which is within the scope of the Journal. Therefore, I recommend the publication of the submitted manuscript in the Journal after minor revision.

Some elements of experimental analysis, interpretation, and presentation require corrections.

 Line 304: Authors should change “America” to “USA”

Please, the authors have to follow the instructions for the authors. In Lines 203-204; 215, 227-234 etc. should use the same font size.

I kindly suggest the authors to transfer Figure 1. A-C to the „Results“.

Also, Figure 2 (Line 79) was given in the text of the manuscript before Figure 1 (Line 363).

Reviewer 3 Report

Manuscript molecules- 1779204 describes identification of 135 chemical components in Cephalanthus tetrandrus (Roxb.) Ridsd. et Badh. F. using UHPLC-Q-Exactive Orbitrap Mass Spectrometry. The manuscript is generally well-written. The following items should be either added or clarified:

1.     lines 27-37:  Since not enough studies have been previously done about the Cephalanthus tetrandrus (Roxb.) Ridsd. et Badh. F (CT), it is very important to add a figure of this plant for clarification reason.

2.     Line 52: it is unqualified to claim the established method as a rapid strategy since a lot of UHPLC-Q-TOF methods for TCM identification purposes are faster than this method (i.e. 30mins).

3.     Reference 2: please correct the sign between the page number.

4.  In section 2.1, the authors majorly indicated the workflow of compounds identification, no qualitative-relevant information was discussed. Please correct the title of section 2.1.

5.     Information the authors talked in section 3.1 should be moved to the section of result.

Author Response

This manuscript is a resubmission of an earlier submission. The following is a list of the peer review reports and author responses from that submission.

Round 1

Reviewer 1 Report

The manuscript discussed  UPLC-MS/MS of Cephalanthus tetrandrus (Roxb.) Ridsd. et Badh. F (CT) belongs to the Rubiaceae family which used in traditional Chinese medicine (TCM) for  treating some diseases.
UPLC-Q-Exactive Orbitrap using "Thermo Scientific Hypersil GOLDTM aQ (100 mm×2.1 mm, 1.9 μm) was used with  mobile phase 0.1% formic 
acid water-acetonitrile, flow rate is 0.3 mL/min, injection volume is 2 μL, 
ESI ion source in positive and negative ion .

Generally, the manuscript needs another part for application or at least prove how these identified compounds have effect on the different diseases which used for treatment in TCM.

For quality control, authors should collect more than one plant sample from different parts.

The authors should at least clarify how these compounds treat mentioned diseases.

there is many phrases needs editing.

For chlorogenic acid derivatives how they are 67 compound?

"Chlorogenic acid is caffeoyl quinic acid"

Reviewer 2 Report

The manuscript presents interesting and valuable work, which is within the scope of the Journal.

- line 12 (Abstract): traditional Chinese medicine (TCM) should delete "TCM" because the abbreviation was not used in the Abstract

- line 42: Authors should change „Thin layer Chromatography“ to „thin layer chromatography“

This paper contains a lot of spelling mistakes (e.g. line 91).

Cis and trans should be written in italic type.

Please, the authors have to follow the instructions for the authors. Lines 203-206 should use the same font size.

Figure 2 text is challenging to follow. Please, correct it.

Please, the authors have to follow the instructions for the authors. The number of images should be correct (e.g. Figure 2: lines 364 - 373 and lines 376 - 379).

This manuscript does not contain part of the discussion, and the scientific contribution is very poorly presented, so it looks like an excellent laboratory report. Unfortunately, a lot of mistakes affect the reading and following of this manuscript.

In my opinion, the proposed manuscript is acceptable as the publication suitable for Molecules after major revision.

Reviewer 3 Report

Tang et al. used the HRMS methods to successfully identify chemical components in Cephalanthus tetrandrus (Roxb.) Ridsd. et Badh. F. This manuscript filled the scientific knowledge gap about CT and provide fundamental information and evidence to support further studies about its medicinal uses. Overall, this manuscript is well designed and organized. Before acceptance of this manuscript for publication, I would suggest the following revisions:

  1. This manuscript should have language editing to improve the clarity of the manuscript and help highlight the research.
  2. The resolution of Figure 1 is low. Please provide the figure of high quality.
  3. Section 1, lines 28-38: The references that the authors cited were not sufficient to support their description of the background of Cephalanthus tetrandrus (Roxb.) Ridsd. et Badh. F (CT). Please cite more relevant literature to support the statements and add a figure of this plant for clarification since there are not many papers talking about this Cephalanthus In addition to this, the authors cited Phillipson and Hemingway’s paper, which talks about the components found in Cephalanthus occidentalis. This is an impropriate citation as the components were identified from the species of C.occidentalis, not CT, please correct it.
  4. Line 47: please use the abb., NMR, instead of the full name since it has been introduced before.
  5. Line 61, please state the full name of PRM.
  6. Throughout the entire manuscript, the authors stated the m/z of analytes with different decimal places. However, high-resolution mass spectrometry is only capable of accurately measuring the m/z of chemical to the 3rd or 4th decimal places, the decimal places after the 3rd or 4th place would be a conjecture made by the system and unable to be reproduced by others. Please correct the numbers throughout the entire manuscript and make sure all numbers have the same decimal places.
  7. Section 4.2: The authors did not state the sample source information as well as the drying and powdering processes of the tested CT. It is well-known that different collecting places would lead the compositions of herbal medicine to be different. And the drying and powdering processes would also potentially impact the stability of some components as well as the extraction rate. All these above points together can make the identification results irreproducible. Please state where the CT was collected from and the manufacturing process of the dried powder if it is known (or the product information).
  8. Section 4.2: Did the authors do any pilot study previously to learn about the extraction yield? In other words, how did the authors know 1g CT in 20mL 70% aqueous ethanol for 1hr could help to fully extract the parts from CT. In addition to this extraction method, please state the temperature of the extraction process.
  9. Section 4.2, line 311, and Section 4.3, line 317: The authors kept the methanolic samples at 4 ℃ for storage but set the autosampler temperature at 10 ℃. Is there any specific reason that supports the authors to store the stock and sample solution in different temperature environments?
  10. Section 4.4 and 4.5: Please cite the corresponding reference(s) to support the authors’ statements about the predictive pathways.